# Carvacrol Encapsulation in Chitosan–Carboxymethylcellulose–Alginate Nanocarriers for Postharvest Tomato Protection

**DOI:** 10.3390/ijms25021104

**Published:** 2024-01-16

**Authors:** Eva Sánchez-Hernández, Alberto Santiago-Aliste, Adriana Correa-Guimarães, Jesús Martín-Gil, Rafael José Gavara-Clemente, Pablo Martín-Ramos

**Affiliations:** 1Department of Agricultural and Forestry Engineering, ETSIIAA, Universidad de Valladolid, 34004 Palencia, Spain; eva.sanchez.hernandez@uva.es (E.S.-H.); alberto.santiago@estudiantes.uva.es (A.S.-A.); adriana.correa@uva.es (A.C.-G.); jesus.martin.gil@uva.es (J.M.-G.); 2Packaging Group, Institute of Agrochemistry and Food Technology (IATA-CSIC), Av. Agustín Escardino, 7, 46980 Paterna, Spain; rgavara@iata.csic.es

**Keywords:** biopolymeric nanoparticles, energy-dispersive X-ray spectroscopy, infrared spectroscopy, nanoencapsulation, natural fungicides, postharvest fruit diseases, stimuli-responsive systems, shelf-life extension, sustainable crop protection, transmission electron microscopy

## Abstract

Advancements in polymer science and nanotechnology hold significant potential for addressing the increasing demands of food security, by enhancing the shelf life, barrier properties, and nutritional quality of harvested fruits and vegetables. In this context, biopolymer-based delivery systems present themselves as a promising strategy for encapsulating bioactive compounds, improving their absorption, stability, and functionality. This study provides an exploration of the synthesis, characterization, and postharvest protection applications of nanocarriers formed through the complexation of chitosan oligomers, carboxymethylcellulose, and alginate in a 2:2:1 molar ratio. This complexation process was facilitated by methacrylic anhydride and sodium tripolyphosphate as cross-linking agents. Characterization techniques employed include transmission electron microscopy, energy-dispersive X-ray spectroscopy, infrared spectroscopy, thermal analysis, and X-ray powder diffraction. The resulting hollow nanospheres, characterized by a monodisperse distribution and a mean diameter of 114 nm, exhibited efficient encapsulation of carvacrol, with a loading capacity of approximately 20%. Their suitability for phytopathogen control was assessed in vitro against three phytopathogens—*Botrytis cinerea*, *Penicillium expansum*, and *Colletotrichum coccodes*—revealing minimum inhibitory concentrations ranging from 23.3 to 31.3 μg·mL^−1^. This indicates a higher activity compared to non-encapsulated conventional fungicides. In ex situ tests for tomato (cv. ‘Daniela’) protection, higher doses (50–100 μg·mL^−1^, depending on the pathogen) were necessary to achieve high protection. Nevertheless, these doses remained practical for real-world applicability. The advantages of safety, coupled with the potential for a multi-target mode of action, further enhance the appeal of these nanocarriers.

## 1. Introduction

The perishability of harvested fruits and vegetables, influenced by environmental factors, storage conditions, and transportation, poses challenges to product quality and shelf life. Extensive efforts have been directed towards alternative coatings, primarily using new edible biopolymers for packaging, acknowledged as generally recognized as safe (GRAS) substances. These edible films and coatings play a pivotal role in preserving fruits and vegetables, addressing the increasing demands of hunger and agricultural management, and enhancing food shelf life, barrier properties, and nutritional attributes. This is achieved by reducing respiration and ripening rates, mitigating ethylene levels, controlling moisture loss, and suppressing microbial activities, as extensively reviewed in the recent literature [1,2].

As polymer science and nanotechnology advance, there is a growing need for novel biopolymer blends endowed with multiple functionalities, especially for applications during storage. Among these functionalities, the development of delivery systems, such as nanocapsules, emerges as a promising avenue for entrapping and enhancing the absorption, stability, and functionality of bioactive compounds (BACs) [3].

Chitosan (CS), cellulose, and alginate (ALG) are extensively studied biopolymers for postharvest fruit protection, either as coatings or components of encapsulation systems [4,5,6,7]. Chitosan, due to its advantageous barrier properties against gases and water vapor, along with antimicrobial properties, stands out for packaging and coating applications [8]. Moreover, chitosan nanoparticles exhibit antimicrobial properties through chelation effects and ionic interactions, inhibiting nutrient transport and inducing cell death [9]. Chitosan oligomers (COS) offer advantages over medium-molecular-weight chitosan in terms of size, solubility, and reactivity.

Carboxymethylcellulose (CMC), a cellulose derivative widely used in various industries, contributes mechanical strength, controllable hydrophilicity, and viscosity. It finds applications in food, paper, textile, pharmaceutical, biomedical engineering, wastewater treatment, energy production, and quality maintenance of agricultural products [10].

Alginate, widely used in biomedical sciences and engineering, is valued for its biocompatibility, non-toxicity, low cost, and ability to form hydrogels when cross-linked with divalent cations [11].

Combining the antibacterial properties of chitosan, the resistance of CMC, and the moisture absorption, permeability, ductility, and film-forming capacity of ALG has led to the exploration of binary or ternary combinations of these components. Interactions between chitosan or COS with CMC, as well as the complexation of chitosan with ALG, have been extensively studied, exploring their oppositely charged polyelectrolyte nature [12,13,14]. Non-covalent crosslinking and hydrogen bonding have also been employed for the formation of CMC/ALG/CS composites [15,16].

In the realm of microcapsules, notable examples include CS–ALG microcapsules using glutaraldehyde [17] (as in that study, CMC did not intervene in the microcapsules, appearing only as a dopant of the colloidal particles of CaCO_3_ used as a template) and ALG–CMC microcapsules crosslinked with glutaraldehyde and copper sulfate [18]. Additionally, CMC–ALG/CS hydrogel beads represent a coating of CMC on ALG and CS beads, not a ternary combination [19]. Despite various combinations, only films [20] or solid beads [16], not hollow micro/nanospheres, have been reported with the concurrent use of all three biopolymers (or for analogous mixtures in which CS was replaced with chitosan biguanidine hydrochloride [15]).

The current understanding of the COS–CMC–ALG system for transporter production underscores the need for alternatives to glutaraldehyde. This study proposes methacrylation of the components, a method previously used for COS–lignin nanocarriers (NCs) [21] and COS-graphitic carbon nitride [22]. Acrylation of ALG [23] and CMC [24], along with the use of sodium tripolyphosphate (STPP), as seen in the CS-STPP-ALG system [25], provides the background for this approach.

The aim of this study is to synthesize, characterize, and apply NCs comprising a ternary complex of COS, CMC, and ALG, utilizing methacrylic anhydride (MA) and STPP as crosslinking agents. Carvacrol, a monoterpenoid phenol widely distributed in the essential oils of aromatic plants such as oregano, thyme, and savory, has been chosen for encapsulation and dispensation. This selection is based on its diverse biological and pharmacological properties, including antimicrobial, antioxidant, and anti-inflammatory effects, as well as its well-established safety profile, being a European Food Safety Authority- and US Food and Drug Administration-approved compound [26,27,28]. Furthermore, this decision is supported by a previous study on a carvacrol-loaded CS nanomaterial [29].

## 2. Results

### 2.1. Morphological Analysis by Transmission Electron Microscopy

The empty COS–CMC–ALG NCs are depicted in Figure 1a,b. The size distribution curve (*n* = 40, Figure 1c) exhibits a single peak, demonstrating a log-normal distribution with an average diameter of 114.35 nm and a standard deviation of 41.03 nm. This results in a polydispersity index (*p*) of 0.36, slightly exceeding the commonly accepted threshold of 0.3 for drug delivery applications using NCs, yet indicating monodispersity [30]. Loading with carvacrol (Figure 1d) did not induce a significant alteration in size (a log-normal distribution with an average diameter of 121.53 ± 43.11 nm, *p* = 0.35), although the shape approached that of perfect spheres.

### 2.2. Bioactive Compound Encapsulation and Release Efficiencies

Regarding encapsulation efficiency, it spanned from 85 to 89%. Concerning release efficiency, initial attempts involving the direct exposure of carvacrol-loaded NCs to the fungi secretome in sealed vials for headspace sampling proved unsuccessful. The minimum inhibitory concentration (MIC) values, discussed below, were minimal, and the released amount of carvacrol, diluted in the headspace, fell below the limit of detection. Instead, a method involving direct exposure of carvacrol-loaded NCs to chitosanase followed by HPLC was employed to obtain a release efficiency (RE) estimation, yielding RE values of up to 90%.

### 2.3. Energy-Dispersive X-ray Analysis

Energy-dispersive X-ray (EDAX) analysis of the empty COS–CMC–ALG NCs (Appendix A) yielded the following atomic percentages: 43.02% in C, 9.59% in N, 40.01% in O, 6.15% in P, and 1.23% in Na. While the C and O percentages were close to the expected ones (43.5 and 41.3%, respectively), the N and P percentages were higher than expected (9.6 vs. 4.5% for N and 6.2 vs. 3.8% for P), and the Na percentage was much lower than expected (1.2 vs. 7%). The variability in N content may be tentatively ascribed to the degree of deacetylation of COS, which typically fluctuates between 5 and 8% [31]. As for the drastic decrease in Na content, it may be regarded as indirect evidence of the cross-linkage of CMC and ALG. Considering that the hydrogen content was excluded from the estimated percentages, the agreement between the obtained and expected atomic percentages can be considered reasonable.

Regarding the EDAX results for the carvacrol-loaded NCs, the atomic percentages were as follows: 47.02% in C, 5.67% in N, 40.69% in O, 5.81% in P, and 0.81% in Na (Appendix A). Assuming an approximate 20% weight percentage of carvacrol (based on an average encapsulation efficiency of 87%), theoretically expected values would be 52.5% in C, 3.6% in N, 35.1% in O, 3.0% in P, and 5.7% in Na. The deviation in N, P, and Na aligns with that mentioned earlier for the empty NCs. Regarding the C and O percentages, the differences were higher than those observed for the empty NCs, but they may be attributed to the semi-quantitative nature of the analysis technique.

### 2.4. X-ray Powder Diffraction Study

The X-ray powder diffraction pattern of the carvacrol-loaded NCs (Appendix A) suggests their low crystallinity. However, a tentative peak assignment can be proposed as follows: the peak at the lowest 2*θ* may be correlated with the peak at 10.4° of COS [32]; the peak at 2*θ* = 13.4° could be attributed to the (110) plane of pure sodium alginate [33]; the peak at 2*θ* = 18° might be assigned to CMC [34] or to the (200) plane of STPP [35]; the peak at 2*θ* = 25.3° may correspond to the (002) plane of CMC [34]; the peak at 2*θ* = 28.5° could be related to CMC [34] or the (402) plane of STPP [35]; the peak at 2*θ* = 43° may be associated with the (223) plane of STPP [35] and perhaps the (311) plane of tripolyphosphate; and the peak at 2*θ* = 44.5° might be linked to the (602) plane of STPP [35].

### 2.5. Infrared Vibrational Study

The primary bands observed in the Fourier Transform Infrared Spectroscopy (FTIR) spectra of both the empty and carvacrol-loaded NCs (Appendix A), along with their corresponding assignments, are outlined in Table 1. Minor differences in the spectra were noted, aligning with prior reports on the successful encapsulation of plant extracts [22]. Bands present in the loaded COS–CMC–ALG NCs and absent in the empty NCs could be attributed to functional groups associated with carvacrol. The presence of these bands, coupled with their weak intensities, suggests that while the majority of the product would be encapsulated, some coating of the outer surface of the NCs with carvacrol may also occur.

### 2.6. Thermal Analysis

In the differential scanning calorimetry (DSC) thermogram of the carvacrol-loaded NCs (Appendix A), four endothermic peaks were observed at 163, 202, 220, and 236 °C. The initial endotherm may be attributed to the glass transition temperature of COS (Tg = 165 °C). The second endotherm, appearing as a shoulder at 202 °C, bears resemblance to an effect observed around 193 °C for COS–TPP beads [36]. The third endotherm, at 220 °C, aligns with an effect previously documented for COS–MA [37]. Regarding the fourth endotherm at 236 °C, it is conceivable that it corresponds to the boiling point of carvacrol (241 °C). Consequently, the thermal profile appears to be influenced by the thermal characteristics of the primary constituents in terms of weight, namely, COS, STPP, carvacrol, and MA.

### 2.7. Antifungal Activity

#### 2.7.1. In Vitro Antifungal Activity

The antifungal susceptibility test results are presented in Figure 2. Carvacrol exhibited higher efficacy against *Botrytis cinerea* Pers. (MIC = 500 μg·mL^−1^) compared to *Colletotrichum coccodes* (Wallr.) S. Hughes and *Penicillium expansum* Link, with MICs of 1000 and 1500 μg·mL^−1^, respectively. Empty NCs displayed considerable antifungal activity, attributed to the antimicrobial properties of COS, with inhibition values ranging from 93.75 to 125 μg·mL^−1^.

Concerning carvacrol-loaded NCs, a clear enhancement of activity was observed in all cases, resulting in MICs as low as 23.34 μg·mL^−1^ for *B. cinerea* and *C. coccodes*, and 31.25 μg·mL^−1^ for *P. expansum*. Table 2 provides the 50% and 90% effective concentrations (EC_50_ and EC_90_, respectively) for different treatments and pathogens, from which synergy factors in the 5.1–9.7 and 7.3–7.9 range may be calculated for the EC_50_ and EC_90_ values, respectively.

#### 2.7.2. Ex Situ Postharvest Protection Tests

Following in vitro results, BAC-loaded NCs were subsequently applied in postharvest protection bioassays on tomato fruits artificially inoculated with each pathogen. Recognizing potential influences on treatment efficacy due to the absorption and metabolism of the phytosanitary treatment by the fruit, as well as variability among fruits, two concentrations, 50 and 100 μg·mL^−1^, were tested.

Figure 3 visually presents external and internal lesions of tomato fruits inoculated with pathogens and their respective positive controls. Concurrently, Table 3 provides a quantitative comparison based on measurements of surface lesions. Notably, the lower treatment dose (50 μg·mL^−1^) proved effective in achieving complete protection in fruits inoculated with *P. expansum*. However, protection against *B. cinerea* necessitated the higher treatment dose (100 μg·mL^−1^). In the case of *C. coccodes*, complete protection was not attained with either dose, but a protection level exceeding 90% was achieved at the concentration of 100 μg·mL^−1^, compared to the positive control, suggesting that a slightly higher dose would be required.

## 3. Discussion

### 3.1. On the COS–CMC–ALG System Assemblage and the Carvacrol Encapsulation Mechanism

In prior studies involving the three system components, such as the solid beads reported by Wang et al. [16], various interactions between CS, CMC, and ALG were suggested, encompassing electrostatic interactions, chelation, and hydrogen bonds. Lan et al. [20], in their work on CS, CMC, and ALG films, also supported the presence of hydrogen bonding. Salama et al. [15], in the context of a CMC/ALG/chitosan biguanidine hydrochloride (CBg) ternary system for edible coatings, indicated that CBg, functioning as a polycationic polymer, readily interacted with negatively charged CMC and ALG. The ionic interactions between C=NH_2_^+^ of CBg and COO^−^ groups of CMC and ALG were complemented by hydrogen bonding interactions among CMC, ALG, and CBg.

The scenario in the current study differs, given that the system assembly was mediated by two cross-linking agents, namely MA and STPP. The use of MA would emulate the effect of N,N′-dicyclohexylcarbodiimide as a cross-linking agent between CS and ALG, as reported by Baysal et al. [38], allowing for covalent binding of the macromolecules. Regarding the role of STPP, intra- and intermolecular linkages would be established between the negatively charged groups of TPP ($P_{3}O_{10}^{5-}$ and $HP_{3}O_{10}^{4-}$) and a fraction of the positively charged amino groups (–NH_3_^+^) of COS [39].

Concerning the encapsulation of BAC (carvacrol), in the work by Martínez-Hernández et al. [40] on carvacrol-loaded CS nanoparticles, the specific type of chemical interaction was not specified. However, it may be hypothesized that hydrogen bonding occurs between TPP and carvacrol (Figure 4).

Hence, the most likely arrangement for the reported COS–ALG–CMC NCs would entail covalent binding (mediated by MA) among the three macromolecules, ionic cross-linking between tripolyphosphate anions (TPP^−^) and protonated amine groups of the COS, along with hydrogen bonding facilitating the encapsulation of BAC.

### 3.2. Comparison of Carvacrol Antimicrobial Activity

The antimicrobial activity of carvacrol is attributed to its interactions with the lipid bilayer of the cytoplasmic membrane, resulting in membrane destabilization, disruption of cell walls, leakage of cell components, and eventual lysis. Carvacrol’s impact extends to adenosine triphosphate synthesis, causing a reduction in energy-dependent processes, inhibiting efflux pumps responsible for antibiotic resistance, disturbing protein synthesis, and altering quorum sensing [41,42].

In the context of its application as a biorational against the three fungi under investigation, it is important to assess its efficacy before encapsulation. A summary of previous studies of carvacrol against *B. cinerea*, *Colletotrichum* spp., and *Penicillium* spp. is presented in Appendix A [43,44,45,46,47,48,49,50,51,52,53,54,55,56,57,58,59,60]. Caution is advised when interpreting these results due to potential variations in isolates (or species, especially in the genus *Colletotrichum* and *Penicillium*) across diverse studies. Additionally, variations in testing methodologies and units used to express results may contribute to discrepancies in comparisons. Moreover, noteworthy differences in effectiveness between essential oils from *Origanum* spp. and *Thymbra spicata* L., with carvacrol as the main component, and commercial carvacrol have been observed, with the activity of pure carvacrol being superior [57,59,60].

For *B. cinerea*, with a MIC of 500 µg·mL^−1^ found for the isolate under study (see Table 2), the efficacy appears lower compared to reports by Tsao and Zhou [44] and Abbaszadeh et al. [45] for commercial carvacrol, demonstrating inhibitory effects at 100 and 300 µg·mL^−1^, respectively.

Regarding *C. coccodes*, with a MIC of 1000 µg·mL^−1^ in this study (Table 2), the efficacy appears higher than that reported by Ochoa-Velasco et al. [49] against *Colletotrichum gloeosporioides* (Penz.) Penz. & Sacc., where inhibition was achieved at 1500 µg·mL^−1^. However, it did not match the effectiveness of carvacrol used by Zhao et al. [51] against *Colletotrichum fructicola* Prihast., L. Cai & K.D. Hyde, which displayed an EC_50_ value of only 31.97 µg·mL^−1^.

Concerning *Penicillium* spp., the carvacrol used against *P. expansum* (MIC = 1500 µg·mL^−1^, Table 2) demonstrated lower efficacy compared to that reported by [45], where MIC values of 150 and 125 µg·mL^−1^ were reported for *Penicillium citrinum* Thom and *Penicillium chrysogenum* Thom, respectively.

### 3.3. Comparison of Efficacy with Conventional Fungicides

Concerning the in vitro activity, Table 4 presents the efficacy values of three conventional fungicides against the targeted plant pathogens. The results, previously reported by our group in prior works, correspond to the same isolates, facilitating direct comparisons. Azoxystrobin exhibited ineffectiveness in inhibiting fungal growth in any of the studied pathogens at the recommended dose of 62,500 µg·mL^−1^, emphasizing the critical need for monitoring and managing fungicide resistance in plant pathogens to ensure effective disease control. Azoxystrobin, a potent quinone inhibitor fungicide, has exhibited high efficacy in addressing a broad spectrum of plant diseases [61]. However, its site-specific mode of action poses a substantial risk of resistance development in phytopathogenic fungal populations. This risk stems from potential alterations in the mitochondrial cytochrome b gene, resulting in variations in the peptide sequence that impede fungicide binding. Such resistance mechanisms to azoxystrobin have been documented in several significant phytopathogenic fungi [62].

Fosetyl-Al demonstrated inhibition of *B. cinerea* and *C. coccodes* at the recommended dose of 2000 µg·mL^−1^, while only achieving 65% inhibition in *P. expansum* at the same dose.

As regards mancozeb, despite successfully inhibiting mycelial growth in the studied pathogens, even at doses 10 times lower than recommended (150 µg·mL^−1^), it was prohibited by the European Commission in 2021 [63]. This decision was prompted by its acknowledged properties as a human carcinogen [64].

**Table 4 ijms-25-01104-t004:** In vitro efficacy of three conventional fungicides against the three fungal pathogens under study at the dose recommended by the manufacturer (Rd) and a tenth of the recommended dose (Rd/10).

Commercial Fungicide	Pathogen	Radial Growth of Mycelium (mm)		Inhibition (%)		Ref. **
		Rd/10	Rd *	Rd/10	Rd *	
Azoxystrobin	*B. cinerea*	12	51	84	32	[65]
	*C. coccodes*	30.6	24.4	59.2	67.5	[66]
	*P. expansum*	38.9	25.6	48.1	65.9	[65]
Mancozeb	*B. cinerea*	0	0	100	100	[65]
	*C. coccodes*	0	0	100	100	[66]
	*P. expansum*	0	0	100	100	[65]
Fosetyl-Al	*B. cinerea*	38	0	49.3	100	[65]
	*C. coccodes*	0	0	100	100	[66]
	*P. expansum*	67.2	26.1	10.4	65.2	[65]

* Rd = 62.5 mg·mL^−1^ of azoxystrobin (250 mg·mL^−1^ for Ortiva^®^, azoxystrobin 25%), 1.5 mg·mL^−1^ of mancozeb (2 mg·mL^−1^ for Vondozeb^®^, mancozeb 75%), and 2 mg·mL^−1^ of fosetyl-Al (2.5 mg·mL^−1^ for Fesil^®^, fosetyl-Al 80%). The control (PDA) exhibited a radial growth of the mycelium of 75 mm. All provided values of mycelial growth are means of three replications. ** Efficacy values against the same isolates previously reported in other studies by our group.

When compared to the three fungicides mentioned, the in vitro antifungal activity of carvacrol-loaded NCs appears highly promising, provided that complete inhibition was achieved at doses below 30 µg·mL^−1^, suggesting a potent antifungal effect.

Regarding ex situ activity, there is a limited body of prior research on nanocarriers involving biopolymers against the studied phytopathogens. Specifically, for *Penicillium* spp., no available reports were identified. However, comparisons can be drawn for the other two fungi, although caution is warranted, considering potential differences arising from the use of distinct isolates or species.

In a study by Machado et al. [67], cellulose-based nanocarriers loaded with commercial hydrophobic fungicides (captan at 2000 and 3000 µg·mL^−1^, pyraclostrobin at 2000 and 3000 µg·mL^−1^, and their mixture) demonstrated inhibition values against *B. cinerea* ranging from 0.5 to 25 µg·mL^−1^ for captan (depending on loading and nanocarrier formulation), in the 5–10 µg·mL^−1^ range for pyraclostrobin, and 10 µg·mL^−1^ for the mixture. Similarly, Lin et al. [68] encapsulated natamycin in zein/carboxymethyl chitosan core–shell nanoparticles, achieving a mycelial inhibition of 64.4% against *B. cinerea* at a concentration of 10 µg·mL^−1^. Additionally, Liu et al. [69] utilized ethyl cellulose polymer microcapsules loaded with fluazinam against gray mold, reporting high inhibition at 0.2 µg·mL^−1^.

For *Colletotrichum* spp., Liang et al. [70] loaded lignin-based nanocarriers with difenoconazole, yielding EC_50_ values in the range of 0.34–0.40 µg·mL^−1^ against *C. gloeosporioides*.

Consequently, the efficacy of carvacrol-loaded nanocarriers, requiring a dose of 100 µg·mL^−1^ for high protection, may be lower than that of conventional fungicides. However, the dose remains sufficiently low for practical real-world applicability. The benefits in terms of safety and the potential for a multi-target mode of action represent additional advantages, possibly offsetting the slightly lower activity. Nevertheless, it is crucial to emphasize the necessity for further testing across various isolates and fungi to validate wide-spectrum activity and rule out the potential for resistance development.

## 4. Materials and Methods

### 4.1. Reagents and Fungal Isolates

High-molecular-weight chitosan (CAS 9012-76-4; 310 to 375 kDa) was sourced from Hangzhou Simit Chem. & Tech. Co. (Hangzhou, China). Neutrase^®^ enzyme was provided by Novozymes A/S (Bagsværd, Denmark). Sodium carboxymethylcellulose (CAS 9004-32-4; USP reference standard), sodium alginate (CAS 9005-38-3; pharmaceutical secondary standard), acetic acid (CAS 64-19-7; purum, 80% in H_2_O), methacrylic anhydride (CAS 760-93-0; ≥94%), sodium tripolyphosphate (CAS 7758-29-4; ≥98%), carvacrol (CAS 499-75-2, 98%), chitosanase from *Streptomyces griseus* (Krainsky) Waksman and Henrici (EC 3.2.1.132, CAS 51570-20-8), methanol (UHPLC, suitable for mass spectrometry, CAS 67-56-1), tetrahydrofuran (THF, CAS 109-99-9; ≥99.9%), and Tween^®^ 20 (CAS 9005-64-5) were procured from Merck (Darmstadt, Germany). Potato dextrose broth (PDB) and potato dextrose agar (PDA) were supplied by Becton, Dickinson, and Company (Franklin Lakes, NJ, USA).

*Botrytis cinerea* (CECT 20973) and *P. expansum* (CECT 20906) were obtained from the Spanish Type Culture Collection (Valencia, Spain), while *C. coccodes* (CRD 246/190) was sourced from the Regional Diagnostic Center of Aldearrubia (Junta de Castilla y León; Castilla y León, Spain).

### 4.2. Synthesis Procedure

Chitosan oligomers were prepared from high-molecular-weight CS according to the procedure described in the study by Santos-Moriano et al. [71], with the modifications indicated in [21]: 20 g of CS was dissolved in 1000 mL of Milli-Q water, adding citric acid under constant stirring at 60 °C and, once dissolution was achieved, 1.7 mL of Neutrase^®^ endoprotease (1.67 g·L^−1^) was added to degrade the polymer chains. The mixture was subjected to ultrasonication at 20 kHz in cycles with sonication of 10 to 15 min interspersed with cycles without sonication of 5 to 10 min to maintain the temperature in the range of 30 to 60 °C. At the end of the process, a solution with a pH in the range of 4 to 6 was obtained with oligomers of molecular weight between 3000 and 6000 Da, and a polydispersity index (*p*) of 1.6, within the usual range reported in the literature [72]. CMC and ALG were used as purchased, without further purification.

The synthesis was carried out using 1.53 g (0.006 mol) of COS, 0.26 g (0.006 mol) of CMC, and 0.11 g (0.003 mol) of ALG. Additionally, 0.39 g (0.015 mol) of MA and 0.92 g (0.015 mol) of STPP were employed as cross-linking agents.

Methacrylation of the COS/CMC/ALG mixture was carried out according to the procedure outlined in [73] with two important modifications: instead of using CS, in this case, COS, CMC, and ALG were used, and, instead of using epichlorohydrin as a crosslinking agent, MA (in THF) was chosen. The mixture was sonicated in 10- to 15 min cycles interspersed with 5- to 10 min non-sonication cycles at a frequency of 20 kHz. The STPP solution was then added dropwise under vigorous stirring. The resulting mixture was further sonicated and kept at a pH between 6 and 7, under stirring, for 24 h. It was then subjected to a centrifugation process and washed with Milli-Q water, yielding the COS–CMC–ALG complex, with an expected molar ratio of 2/2/1 (2/2/1/5/5 if the cross-linking agents are considered).

### 4.3. Bioactive Compound Encapsulation and Release

Regarding the inclusion of the BAC in the product synthesized as detailed above, the chosen compound for encapsulation was carvacrol. To form a complex akin to inclusion complexes, carvacrol (0.008 mol, 1.2 g), initially dissolved in a methanolic medium, was introduced into the COS/CMC/ALG/MA/STPP solution, followed by sonication and stirring. The subsequent steps of the procedure closely followed those outlined in the preceding section.

To assess the encapsulation efficiency, Fischer et al.’s indirect method [74] was selected. The sample underwent centrifugation at 10,000 rpm for 1 h, and the resulting supernatant, containing the non-encapsulated carvacrol, underwent freeze-drying, was redissolved in methanol, filtered through a 0.2 μm filter, and subjected to analysis via high-pressure liquid chromatography (HPLC) using an Agilent 1200 series system (Agilent Technologies; Santa Clara, CA, USA). Operating conditions replicated those specified in [75], with detection at 274 nm. The EE was computed as EE (%) = [(m_carvacrol initial_ − m_carvacrol supernatant_)/m_carvacrol initial_] × 100, and the reported EE value was an average of 10 repetitions.

The release efficiency was estimated following a method similar to that employed in [65]. This involved introducing a weighed amount of freeze-dried carvacrol-loaded NCs (obtained from the encapsulation efficiency test) and 2.5 U of chitosanase (EC 3.2.1.132) into a methanol/water (1:1, *v*/*v*) solution with light stirring (150 rpm) in the dark for 2 h. An aliquot was sampled, and the released carvacrol underwent the same methodology as explained earlier to determine the residual (non-encapsulated) carvacrol. The release efficiency was calculated as the percentage of the released carvacrol relative to the total amount of carvacrol encapsulated in the NCs across 10 repetitions.

### 4.4. Nanocarriers Characterization

Transmission electron microscopy was used to characterize the NCs’ morphology. A JEOL (Akishima, Tokyo, Japan) JEM 1011 HR microscope (operating conditions: 100 kV; 25,000–120,000 magnification) and a GATAN ES1000W CCD camera (4000 × 2672 pixels) were employed for this purpose. The samples were negatively stained with uranyl acetate (2%). The polydispersity index was computed from TEM data using the formula *p* = *σ*/R_avg_, where *σ* is the standard deviation of the radius in a batch of NCs, and R_avg_ is the average radius of the NCs [76].

The elemental composition of the NCs, before and after loading with carvacrol, was determined by scanning electron microscopy with energy-dispersive X-ray spectroscopy (SEM–EDAX) using an EVO HD 25 (Carl Zeiss; Oberkochen, Germany) instrument. The standardless ZAF quantification method was applied for the analysis.

The phase composition of the carvacrol-loaded NCs was investigated using a D8 Advance diffractometer (Bruker; Billerica, MA, USA) with a Cu K*α* X-ray source (λ = 0.15406 nm). The X-ray powder diffraction pattern was obtained in the 2*θ* = 5–70° range.

Infrared spectra were acquired using a Thermo Scientific (Waltham, MA, USA) Nicolet iS50 FTIR spectrometer with an integrated attenuated total reflection system. The spectra were recorded over the 400–4000 cm^−1^ wavenumber range with a resolution of 1 cm^−1^.

Thermal analysis of carvacrol-loaded NCs (5 mg) was performed with a TA Instruments DSC Q2000 (TA Instruments Inc., New Castle, DE, USA). The instrument’s temperature was calibrated with indium. Nitrogen was used as a purge gas at a flow rate of 50 mL·min^−1^, and the heating rate was 10 °C·min^−1^.

### 4.5. Antifungal Activity Assessment

#### 4.5.1. In Vitro Antifungal Activity

The assessment of antifungal activity for COS–CMC–ALG NCs, both pre- and post-carvacrol encapsulation, as well as non-encapsulated carvacrol (for comparative analysis), employed the poisoned food method following EUCAST antifungal susceptibility testing standard procedures [77]. Solutions of carvacrol and NCs, whether empty or loaded with carvacrol, were incorporated into a PDA medium to achieve concentrations ranging from 7.81 to 1500 μg·mL^−1^. Fungal mycelium plugs from the margins of 1-week-old PDA cultures of *B. cinerea* and *C. coccodes*, as well as from a 2-week-old culture of *P. expansum*, were transferred to plates containing the specified concentrations for each treatment. Different time periods were employed for different fungi, considering that each fungus has its own growth rate. Three plates per treatment/concentration combination were prepared, with two replicates each. Incubation was conducted under specific conditions for each fungus: *B. cinerea* and *C. coccodes* plates were incubated at 25 °C in the dark for 1 week, and *P. expansum* for 2 weeks. The untreated control consisted of pure PDA medium. Radial mycelium growth was evaluated by measuring the average of two colony diameters perpendicular to each other for each repetition. Growth inhibition was determined using the formula ((d_c_ − d_t_)/d_c_) × 100, where d_c_ and d_t_ represent the mean colony diameter of the untreated control and the treated fungus, respectively. Effective concentrations were estimated by fitting a four-parameter logistic equation (dose–response curve). The level of interaction (i.e., synergy factors) was determined according to Wadley’s method [78].

#### 4.5.2. Preparation of Fungal Conidial Suspension

Fungal conidial suspensions were prepared following the procedures outlined in [65,66]. Conidia were harvested from 1-week-old PDB cultures of *B. cinerea* and *C. coccodes*, as well as from a 2-week-old PDB culture of *P. expansum*. After filtration through two layers of sterile muslin to eliminate somatic mycelia, the spore concentration was determined using a hemocytometer and adjusted to 1 × 10^6^ spores (conidia)·mL^−1^.

#### 4.5.3. Ex Situ Protection of Tomato Fruits

The efficacy of COS–CMC–ALG NCs loaded with carvacrol for postharvest protection of tomato fruits (cv. ‘Daniela’), cultivated under EU organic farming regulations at Huerta El Gurullo (Cuevas del Almanzora, Almería, Spain), was assessed as outlined in [65,66]. All tested fruits, exhibiting uniform dimensions (approximately 75 mm in diameter), displayed no apparent signs of disease. Initially, the tomatoes underwent a 2 min surface disinfection using a 3% NaOCl solution. Subsequently, they underwent three rinses with sterile distilled water and were dried on sterile absorbent paper within a laminar flow hood.

The fruits were categorized into four groups: two groups received treatments with COS–CMC–ALG NCs loaded with carvacrol at varying concentrations (50 and 100 μg·mL^−1^, supplemented with 0.2% Tween^®^ 20). The remaining groups functioned as negative (untreated and pathogen-free) and positive (pathogen-infected without treatment) controls.

In aseptic conditions, each fruit was punctured at three equidistant points in the equatorial region using a truncated needle (3 mm diameter × 5 mm depth). The treated fruits were initially injected with 20 µL of the corresponding treatment at each puncture point. After one hour, wounds were inoculated with 20 µL of a fungal spore suspension (1 × 10^6^ conidia·mL^−1^). Positive controls were solely subjected to fungal spore suspension, while negative controls were inoculated with sterile deionized water containing 0.2% Tween^®^ 20.

Each fruit was individually placed in a clean container corresponding to its treatment and pathogen, with sterile moistened cotton, and then incubated at 25 °C for seven days. Lesion diameters were measured twice at right angles to each other on the fruit surfaces. The percentage of lesion size reduction compared to the positive control (0% reduction) was calculated using the formula LSR (%) = [(LS_c_ − LS_t_)/LS_c_] × 100, where LS_c_ represents the lesion diameter of the positive control, and LS_t_ represents the lesion diameter of the treated fruits. On day 7, at the conclusion of the experiment, the tomatoes were incised to analyze the internal lesions.

Notably, a contrast fungicide was not employed in these experiments. This decision aligns with the current Spanish national legislation on the registration of phytosanitary products, which does not presently authorize a fungicide for direct use on postharvest tomatoes.

### 4.6. Statistical Analysis

The mycelial growth inhibition results for the BAC (carvacrol), the empty NCs, and the loaded NCs were assessed using IBM SPSS Statistics v.25 (IBM; Armonk, NY, USA) via the Kruskal–Wallis non-parametric test, contingent upon the non-satisfaction of normality and homoscedasticity requirements, which were evaluated using the Shapiro–Wilk and Levene tests, respectively. Post hoc multiple pairwise comparisons were conducted using the Conover–Iman test.

## 5. Conclusions

The synergistic integration of the antibacterial properties of COS, the resistance of CMC, and the moisture absorption, permeability, ductility, and film-forming capacity of ALG has prompted investigations into their combined applications. The ternary system studied here employed COS, CMC, and ALG in a 2:2:1 ratio. This formulation utilized MA for covalent binding among the three macromolecules and STPP for ionic cross-linking between TPP^−^ and $NH_{3}^{+}$ groups of COS. STPP also facilitates the encapsulation of BACs via hydrogen bonding. The resulting hollow nanospheres exhibited a log-normal distribution, with a diameter of 114 ± 41 nm and a polydispersity index of 0.36. Regarding their efficacy in the vehiculization and release of BACs, an encapsulation efficiency of approximately 87% was achieved for carvacrol, indicating a loading ratio close to 20%. Upon direct exposure to a chitosanase, a release efficiency of up to 90% was recorded. Thermal and vibrational data further supported the successful encapsulation process. The developed encapsulation system was evaluated in vitro against three fruit spoilage fungi, observing MIC values of 23, 23, and 31 μg·mL^−1^ against *B. cinerea*, *C. coccodes*, and *P. expansum*, respectively. These values were significantly lower than those obtained for unencapsulated carvacrol or empty NCs. In postharvest protection tests for tomatoes, a dose of 50 μg·mL^−1^ was required for full protection against *P. expansum*, while a concentration of 100 μg·mL^−1^ fully inhibited *B. cinerea* and almost inhibited *C. coccodes* (reaching a protection level exceeding 90%). This efficacy surpassed that of three non-encapsulated conventional fungicides (azoxystrobin, fosetyl-Al, and mancozeb) but fell below that of other NC-encapsulated fungicides (captan, pyraclostrobin, and fluazinam). Given the sufficiently low dose for practical real-world applicability and the potential benefits associated with the use of generally recognized-as-safe products, this reported material warrants further exploration for postharvest crop protection and other applications in pharmaceuticals, cosmetics, or pre-harvest crop protection contexts.

## Figures and Tables

**Figure 1 ijms-25-01104-f001:**
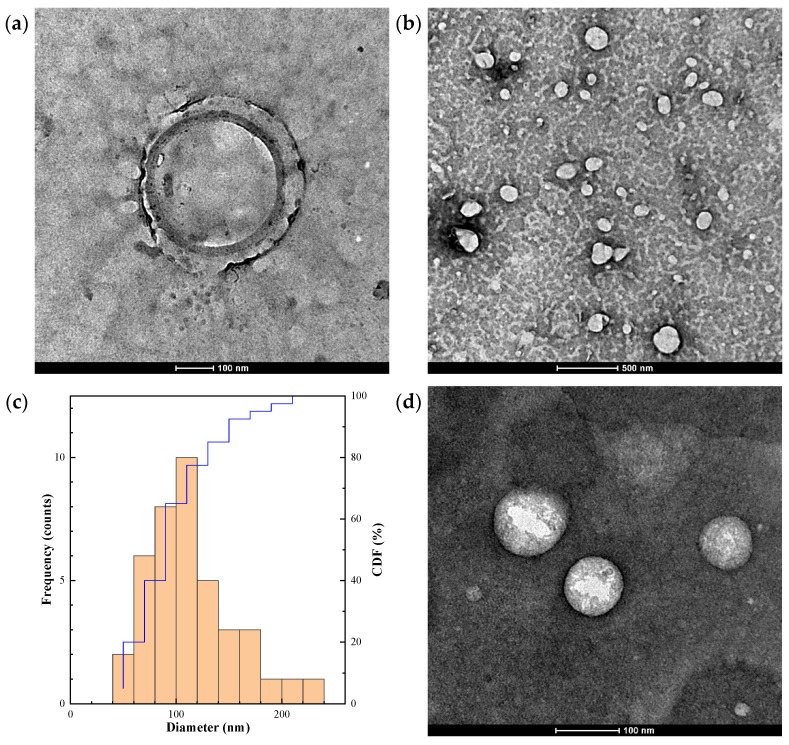
(**a,b**) Transmission electron microscopy (TEM) micrographs of the empty chitosan oligomers–carboxymethylcellulose–alginate (COS–CMC–ALG) nanocarriers (NCs); (**c**) size distribution histogram of the NCs, indicating the frequency and the cumulative distribution function (CDF), represented in blue; (**d**) COS–CMC–ALG NCs loaded with carvacrol.

**Figure 2 ijms-25-01104-f002:**
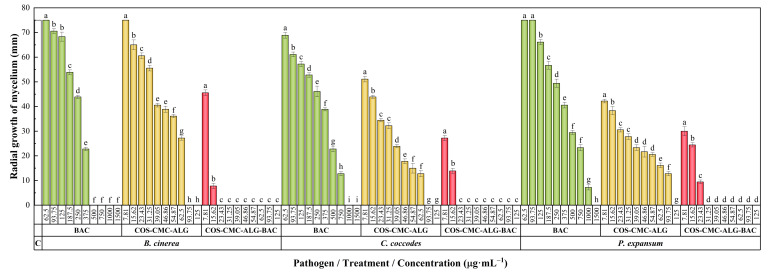
Mycelial growth inhibition achieved with the non-encapsulated bioactive compound (BAC), namely carvacrol, the empty COS–CMC–ALG NCs, and the COS–CMC–ALG NCs loaded with the BAC against the three fungal pathogens under study, namely *B. cinerea*, *C. coccodes*, and *P. expansum*, at concentrations ranging from 7.81 to 1500 μg·mL^−1^. The same letters denote non-significant differences at *p* < 0.05. ‘C’ represents the untreated control (each fungus growing in potato dextrose agar medium with only the extraction solvent added).

**Figure 3 ijms-25-01104-f003:**
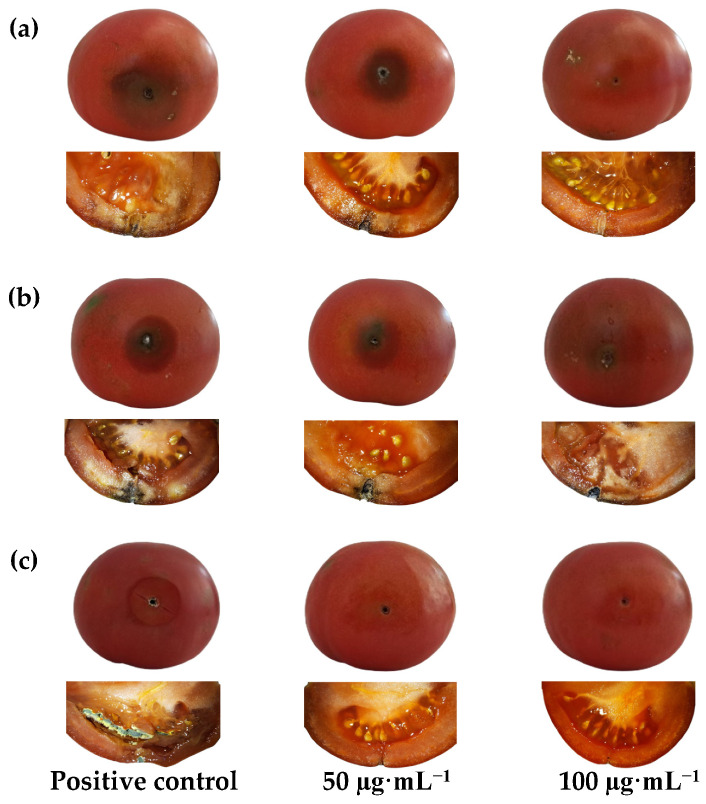
External and internal lesions caused by (**a**) *B. cinerea*, (**b**) *C. coccodes*, and (**c**) *P. expansum* on tomato cv. ‘Daniela’ seven days after artificial inoculation in the presence/absence of COS–CMC–ALG NCs loaded with the BAC (carvacrol) at 50 and 100 µg·mL^−1^.

**Figure 4 ijms-25-01104-f004:**
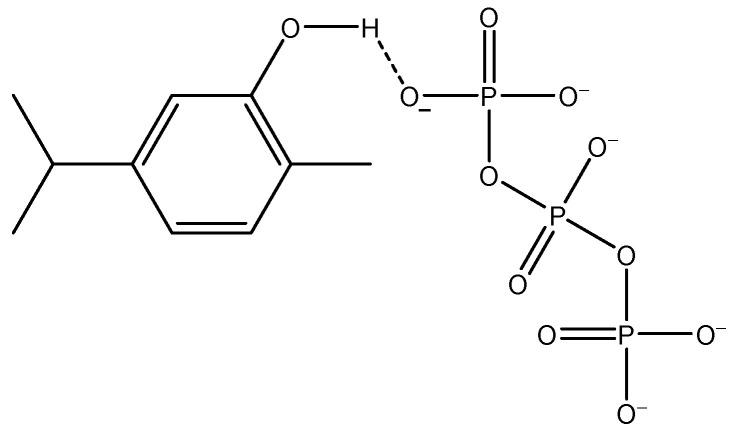
Hydrogen bonding between tripolyphosphate and carvacrol.

**Table 1 ijms-25-01104-t001:** Main bands in the infrared spectrum of the carvacrol-loaded COS–CMC–ALG NCs.

Wavenumber (cm^−1^)		Assignment	Component
Empty NCs	Loaded NCs		
	3350	H–bonded OH stretching	carvacrol
3265	3266	asymmetrical stretching of the –NH group	COS
	2960	CH stretching branched alkane	carvacrol
	1704	C=O asymmetric stretching of phenolic acids	carvacrol
1632	1628	overlapping stretching of alkenes (C=C) and carbonyl (C=O)	COS, CMC
1536	1538	N–H bending of N–acetylated residues (amide II) (after binding to alginate); asymmetrical stretching of COO– groups	COS, CMC
1412	1418	N–H stretching (amide and ether bonds); symmetrical stretching of COO^–^ groups	COS, ALG, CMC
1381	1381	N–H stretching (amide III band)	COS
1248	1250	C–O–C stretching	carvacrol
1149	1149	symmetric and antisymmetric stretching in the PO_4_ group	TPP
1065	1065	–C–O–C bonds	COS
1015	1028	C–O–C stretching; C–O stretching attributed to the saccharide structure	ALG, COS
947	942	C–O stretching vibration of uronic acid residues	ALG
	871	aromatic rings	carvacrol
	810	typical of *p*-substituted aromatic rings	carvacrol

**Table 2 ijms-25-01104-t002:** Effective concentration (EC) values (in µg·mL^−1^) against the fungal pathogens under study obtained for the non-encapsulated BAC (carvacrol), the empty COS–CMC–ALG NCs, and the COS–CMC–ALG NCs loaded with the BAC.

Treatment	EC	*B. cinerea*	*C. coccodes*	*P. expansum*
BAC	EC_50_	282.4	369.8	406.6
	EC_90_	456.2	823.7	1122.7
COS–CMC–ALG	EC_50_	50.8	22.7	17.0
	EC_90_	84.6	73.1	104.7
COS–CMC–ALG–BAC	EC_50_	8.9	5.3	6.4
	EC_90_	18.0	18.3	25.6

**Table 3 ijms-25-01104-t003:** Lesion diameter (LD) and lesion size reduction (LSR) in the presence/absence of COS–CMC–ALG NCs loaded with the BAC (carvacrol) at 50 and 100 µg·mL^−1^ on tomato fruits. Negative control refers to untreated fruit without pathogen, and positive control indicates pathogen-inoculated fruit without treatment.

Pathogen	Treatment	LD (mm)	LSR (%)
−	Negative control	0	100
*B. cinerea*	Positive control	25.4 ± 2.8	0
	BAC-loaded NCs at 50 µg·mL^−1^	17.3 ± 2.3	31.9
	BAC-loaded Ns at 100 µg·mL^−1^	0	100
*C. coccodes*	Positive control	24.2 ± 3.7	0
	BAC-loaded NCs at 50 µg·mL^−1^	21.5 ± 1.6	78.8
	BAC-loaded NCs at 100 µg·mL^−1^	1.8 ± 1.1	92.6
*P. expansum*	Positive control	16.7 ± 1.2	0
	BAC-loaded NCs at 50 µg·mL^−1^	0	100
	BAC-loaded NCs at 100 µg·mL^−1^	0	100

## Data Availability

The data supporting the findings of this study are available within the article and its Appendix A.

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
