# Peer review of "Carvacrol Encapsulation in Chitosan–Carboxymethylcellulose–Alginate Nanocarriers for Postharvest Tomato Protection"

_ijms, 2024, doi:10.3390/ijms25021104_

Round 1

Reviewer 1 Report

Comments and Suggestions for Authors

In the research article entitled “Carvacrol Encapsulation in Chitosan−Carboxymethylcellulose− Alginate Nanocarriers for Postharvest Tomato Protection” the authors characterized carriers formed through the complexation of chitosan oligomers, carboxymethylcellulose, and alginate; regarding the postharvest protection in tomato. This manuscript shows interesting and original results and evaluated safer alternatives to manage postharvest decays in tomato.

However, there are several points that should be carefully taken into consideration.

Here the authors made a comparison with conventional fungicides. As the author only made a literature revision and comparison regarding previous publications, without own results, that comparison described in the discussion section should be rewrote, shortened, and clearly specified, as written, it could confuse the reader.

As for the size of the carriers, they should be considered as microcarriers?

Being Carvacrol the main component, please expand on its nature and origin in the introduction section.

Figure 1: Y-axis title should be written in English. Please give a title to the X-axis.

Line 246: Please introduce the correct citation and remove the bold sentence.

Line 264: Please move Table 4 to supplementary section.

Line 299: Please include the corresponding citation.

Line 482: Please indicate where you injected the corresponding treatment on the tomato.

Reviewer 2 Report

Comments and Suggestions for Authors

Referee report

Carvacrol Encapsulation in Chitosan−Carboxymethylcellulose− Alginate Nanocarriers for Postharvest Tomato Protection, Eva Sánchez-Hernández, Alberto Santiago-Aliste, Adriana Correa-Guimarães, Jesús Martín-Gil, Rafael José Gavara-Clemente and Pablo Martín-Ramos

Dear Editor

In my opinion manuscript is an interesting, but I recommend a revision. All the methodological and editorial errors I find are marked in yellow in the text.

Below my proposition of changes:

All keywords should be specified, e.g. instead of the general word characterization, please list the most important methods used for characterization. There is nothing here about the methods used for protection.

Line 51

The ingredients have their abbreviations, but cellulose is not there, why? Please standardize, e.g. C

Line 55

Please precise

Quasi-zero charge?

Line 65

Ability to gel

In my opinion it is oversimplification

Ability to gel forming or ability to geliation....?

Part 2.4

Please insert the correct degree symbol without this weird underline.

Line 163

The correct nomenclature notation .....p must be written in italics.

Line 180

Should be italic written

Line 242

Should be space

Line 246

Error???

Line 279

Unit

Table 4

Better

This paper or this manuscript

Line 379

Please add initials to distinguish these two people.

Line 402-403

Please insert square brackets in the equation to avoid these double brackets.

Part 4.4

The title is too laconic, please expand.

Line 450

Explain the time differences and why the same time was not used?

Line 487

Is there any correlation between the time assumed here and the previously declared time?

Line 518-522

Style?

Correct the grammar in this text fragment.

Line 628

Should be italic. The names of fungi, enzymes, in vitro, in vivo etc., according to systematic nomenclature, should be written in italics.

Conclusions should be the best written fragment of the manuscript. Authors, those who will potentially quote us, usually read the abstract and conclusions, which is why these parts are so important and must be written perfectly.

General attention to all manuscript.

The manuscript should compare and contrast the ideas in the reviewed literature; and deal with the limitation of the ideas discussed. I have marked in yellow relatively old manuscripts that can be replaced with newer ones. Authors can use these manuscripts as examples:

The influence of polysaccharides/TiO2 on the model membranes of dipalmitoylphosphatidylglycerol and bacterial lipids, Molecules, 27(2) 2022, 343

Edible films made from blends of gelatin and polysaccharide-based emulsifiers - A comparative study, Food Hydrocolloids 96 (2019) 555-567

The effect of chitosan/TiO2/hyaluronic acid subphase on the behaviour of 1,2-dioleoyl-sn-glycero-3-phosphocholine membrane, Biomaterials Advances, 138 (2022), 212934,

Based on my comments, I propose a minor revision.

Comments on the Quality of English Language

Minor revision
